

# Male sex pheromone components in *Heliconius* butterflies released by the androconia affect female choice

Kathy Darragh[1,2,*], Sohini Vanjari[1,2,*], Florian Mann[3,*],
Maria F. Gonzalez-Rojas[4,*], Colin R. Morrison[2,5], Camilo Salazar[4],
Carolina Pardo-Diaz[4], Richard M. Merrill[1,2,6], W. Owen McMillan[2],
Stefan Schulz[3] and Chris D. Jiggins[1,2]

[1] Department of Zoology, University of Cambridge, Cambridge, Cambridgeshire, United Kingdom
[2] Smithsonian Tropical Research Institute, Panama
[3] Institute of Organic Chemistry, Technische Universität Braunschweig, Braunschweig, Germany
[4] Biology Program, Faculty of Natural Sciences and Mathematics, Universidad del Rosario, Bogota, Colombia
[5] Department of Integrative Biology, University of Texas at Austin, Austin, TX, United States of America
[6] Division of Evolutionary Biology, Faculty of Biology, Ludwig-Maximilians-Universität München, Munich, Germany
[*] These authors contributed equally to this work.

Corresponding author
Chris D. Jiggins,
c.jiggins@zoo.cam.ac.uk

## ABSTRACT

Sex-specific pheromones are known to play an important role in butterfly courtship, and may influence both individual reproductive success and reproductive isolation between species. Extensive ecological, behavioural and genetic studies of *Heliconius* butterflies have made a substantial contribution to our understanding of speciation. Male pheromones, although long suspected to play an important role, have received relatively little attention in this genus. Here, we combine morphological, chemical and behavioural analyses of male pheromones in the Neotropical butterfly *Heliconius melpomene*. First, we identify putative androconia that are specialized brush-like scales that lie within the shiny grey region of the male hindwing. We then describe putative male sex pheromone compounds, which are largely confined to the androconial region of the hindwing of mature males, but are absent in immature males and females. Finally, behavioural choice experiments reveal that females of *H. melpomene*, *H. erato* and *H. timareta* strongly discriminate against conspecific males which have their androconial region experimentally blocked. As well as demonstrating the importance of chemical signalling for female mate choice in *Heliconius* butterflies, the results describe structures involved in release of the pheromone and a list of potential male sex pheromone compounds.

## INTRODUCTION

Sex pheromones are species-specific blends of chemical compounds that mediate intraspecific communication between males and females (*Wyatt, 2003*; *Wyatt, 2014*). Among insects, pheromone communication can involve a single chemical, but often

relies on a complex combination of multiple chemical components (*Grillet, Dartevelle & Ferveur, 2006*; *Nieberding et al., 2008*; *Symonds, Johnson & Elgar, 2012*). This chemical complexity provides the potential to convey sophisticated information, such as the inbreeding status of the emitter (*Ando, Inomata & Yamamoto, 2004*; *Van Bergen et al., 2013*; *Menzel, Radke & Foitzik, 2016*), mate quality (*Dussourd et al., 1991*; *Ruther et al., 2009*), and species identity (*Danci et al., 2006*; *Saveer et al., 2014*). Perhaps the best studied insect sex pheromones are those produced by female moths to attract mating partners, often over long distances (*Löfstedt, 1993*; *Smadja & Butlin, 2008*). However, male insects also produce sex pheromones (*Eggert & Müller, 1997*; *Kock, Ruther & Sauer, 2007*; *Ruther et al., 2009*; *Meinwald, Meinwald & Mazzocchi, 1969*), and chemical signalling can occur over short distances (*Nishida et al., 1996*; *Mas & Jallon, 2005*; *Smadja & Butlin, 2008*; *Wicker-Thomas, 2011*; *Grillet et al., 2012*).

Sex pheromones can play a key role in determining the reproductive success of individuals within a species, and may also result in reproductive isolation between species if signals diverge (*Johansson & Jones, 2007*; *Smadja & Butlin, 2008*; *Wyatt, 2014*). Within Lepidoptera, the importance of chemical signalling in mate choice and speciation is well established among moth species (*Phelan & Baker, 1987*; *Löfstedt, 1993*; *Bethenod et al., 2004*; *Dopman, Robbins & Seaman, 2010*; *Lassance et al., 2010*; *Saveer et al., 2014*). Most moths fly at night, when visual signalling is unlikely to be as effective in attracting mates. In contrast, butterflies are mostly diurnal and visual signals are usually important for initial mate attraction (*Vane-Wright & Boppré, 1993*). However, chemical signals can play other roles in butterfly mate choice, with evidence that close-range courtship interactions often involve pheromones emitted by males, in contrast to the long-distance signalling with female-emitted pheromones more commonly observed in moths (*Vane-Wright & Boppré, 1993*). Acceptance behaviour in the queen butterfly *Danaus berenice*, for example, is regulated by a dihydropyrrolizine alkaloid released by the male during courtship (*Brower & Jones, 1965*; *Meinwald, Meinwald & Mazzocchi, 1969*; *Pliske & Eisner, 1969*). Another danaine butterfly, *Idea leuconoe*, displays brush-like structures, called 'hair-pencils', emitting a mixture of volatiles during courtship, which when applied to dummy males elicits an acceptance posture in females (*Nishida et al., 1996*). *Pieris rapae* and *P. brassicae* both use macrocyclic lactones as a pheromone to induce acceptance in females (*Yildizhan et al., 2009*). Finally, in *Bicyclus anynana* males with reduced amounts of male sex pheromone have decreased mating success, implying a direct involvement in reproductive fitness (*Nieberding et al., 2008*; *Nieberding et al., 2012*).

Here we focus on the potential role of male pheromones in *Heliconius* butterflies. *Heliconius* is a diverse Neotropical genus, which has been extensively studied in the context of adaptation and speciation (*Jiggins, 2008*; *Supple et al., 2014*; *Merrill et al., 2015*). These butterflies are well known for Müllerian mimicry, in which unrelated species converge on the same warning signal to more efficiently advertise their unpalatability to predators. Closely related *Heliconius* taxa, however, often differ in colour pattern and divergent selection acting on warning patterns is believed to play an important role in speciation (*Bates, 1862*; *Jiggins et al., 2001*; *Merrill et al., 2011b*).
Male *Heliconius* display conspicuous courtship behaviours likely because the availability of receptive females in nature is limited. Female re-mating is a rare event in *Heliconius* (*Walters et al., 2012*) and males must compete to find virgin females within a visually complex environment (*Merrill et al., 2015*). In addition, males donate a nutrient-rich spermatophore during mating (*Boggs & Gilbert, 1979*; *Boggs, 1981*) which, together with costs associated with extended copulation, will select for discrimination against less suitable mates in both sexes. A combination of colour (hue) and movement stimulates courtship by *Heliconius* males (*Crane, 1955*). More recently, it has repeatedly been shown across multiple *Heliconius* species that males are more attracted to their own warning pattern than that of closely related taxa (*Jiggins et al., 2001*; *Jiggins, Estrada & Rodrigues, 2004*; *Kronforst et al., 2006*; *Melo et al., 2009*; *Muñoz et al., 2010*; *Merrill et al., 2011a*; *Merrill et al., 2011b*; *Merrill, Chia & Nadeau, 2014*; *Finkbeiner, Briscoe & Reed, 2014*; *Sánchez et al., 2015*).

In addition to colour pattern, male *Heliconius* also use chemical signals to locate and determine the suitability of potential mates. This includes the use of green leaf volatiles during mate searching. Six-carbon alcohols and acetates are released by host plants in larger amounts after leaf tissue damage caused by caterpillars, which adult males of the pupal mating species *H. charithonia* then use to find potential mates (*Estrada & Gilbert, 2010*). Once males find pupae they also use chemical cues to determine sex (*Estrada et al., 2010*). Supporting a further role for chemical signals, *Heliconius erato* males distinguish between wings dissected from conspecific and local *H. melpomene* females that are virtually identical in wing pattern, but this effect disappears after wings have been washed in hexane (*Estrada & Jiggins, 2008*).

As well as attraction, chemicals can also be involved in repulsion. Males are repelled by a strong odour released by previously mated females (*Gilbert, 1976*). This 'anti-aphrodisiac' is produced by males soon after eclosion and is then transferred during copulation (*Schulz et al., 2008*). The abdominal glands of male *H. melpomene*, for example, contain a complex chemical bouquet consisting of the volatile compound $(E)$-$\beta$-ocimene together with some trace components and esters of common $C_{16}$—and $C_{18}$—fatty acids with alcohols, where $\beta$-ocimene acts as the main antiaphrodisiac component (*Schulz et al., 2008*). This antiaphrodisiac effect occurs in several *Heliconius* species, which show species-specific patterns of scent gland constituents (*Gilbert, 1976*; *Estrada et al., 2011*).

Despite the focus on male mate choice, analysis of courtship in *Heliconius* has shown that females can exhibit rejection behaviours, such as raising their abdomen and flattening their wings (*Mallet, 1986*; *Klein & Araújo, 2010*). There are also a number of observations that indicate a role for chemical recognition in female mate choice. *Heliconius erato* males separate their wings during courtship to reveal the silvery overlap region, suggested to be involved in the distribution of pheromones. This behaviour, described as androconial exposition, occurs in every courtship that results in mating, suggesting that pheromones influence the female response (*Klein & Araújo, 2010*). Additionally, direct evidence that *Heliconius* females use chemical signals to distinguish conspecific males comes from studies of the closely related species *H. timareta* and *H. melpomene*, which share the same warning patterns in Peru (*Mérot et al., 2015*). Males experimentally treated with abdominal scent gland and wing extracts of heterospecifics show a reduced probability of mating. Chemical
analysis of both abdominal glands and whole wings provides evidence for qualitative and quantitative differences in the chemical signatures between these closely related species (*Mérot et al., 2015*).

Here, we investigate the role of chemical signalling in female mate choice in *Heliconius* at three levels. First, we investigate morphological structures potentially associated with pheromone production. In butterflies, a variety of species-specific structures including brushes, fans, and differentiated scales on wings, legs or abdomen are used to expose pheromones produced in associated glands (*Wyatt, 2003*; *Nieberding et al., 2008*). In particular, male-specific scent glands, termed androconia, are common across the Lepidoptera. In male *Heliconius*, a patch of shiny grey scales is present on the overlapping region of the hind and forewing (Fig. 1). The observed sexual dimorphism in this trait suggests that these are androconia, and may be associated with a male sex pheromone (*Emsley, 1963*). Furthermore, earlier authors have identified brush-like scales in the hindwing androconial region that are the putative site for pheromone production and emission (*Müller, 1912*; *Barth, 1952*). Here we investigate the structure of these scales using scanning electron microscopy. Second, we complement recently published chemical analysis of whole *H. melpomene* wings (*Mérot et al., 2015*) by dissecting wing regions to identify those associated with the production of compounds and identify the potential male sex pheromone compounds isolated from this region. As *H. melpomene rosina* was available for more extensive behavioural experiments at the insectaries in Panama, we focused further work on this population, including repeating chemical analyses to ensure that the sexual dimorphism was also present. Finally, we carry out mate choice experiments in *H. melpomene rosina*, *H. melpomene malleti*, *H. timareta florencia* and *H. erato demophoon* to test the importance of pheromones for female choice in *Heliconius*.

## METHODS

Individuals used for morphological and chemical analyses were from an outbred stock of *Heliconius melpomene plesseni* and *Heliconius melpomene malleti* (sold as *H. m. aglaope*), maintained at the University of Cambridge insectaries (Fig. 1A). These two races are from the region of a hybrid zone in the eastern Andes of Ecuador, and showed considerable inter-racial hybridization in the stocks, so are treated here as a single population and referred to as the Ecuador samples. These stocks were established from individuals obtained from a commercial breeder (Stratford-Upon-Avon Butterfly Farm, Swans Nest, Stratford-Upon-Avon, UK: http://www.butterflyfarm.co.uk). Laboratory stocks were maintained on the larval food plants, *Passiflora menispermifolia* and *P. biflora*. Adult butterflies were fed on ∼10% sucrose solution mixed with an amino acid supplement (Critical Care Formula®; Vetark Professional, Winchester, UK). Further chemical and behavioural analysis were carried out on the mimetic but distantly related *H. melpomene rosina* and *H. erato demophoon* reared at the Smithsonian Tropical Research Institute (STRI) facilities in Gamboa, Panama, and are referred to as the Panama samples. Both males and females of the Panama samples were from outbred stocks established from wild individuals collected in Gamboa (9°7.4′N, 79°42.2′W, elevation 60 m) within the nearby
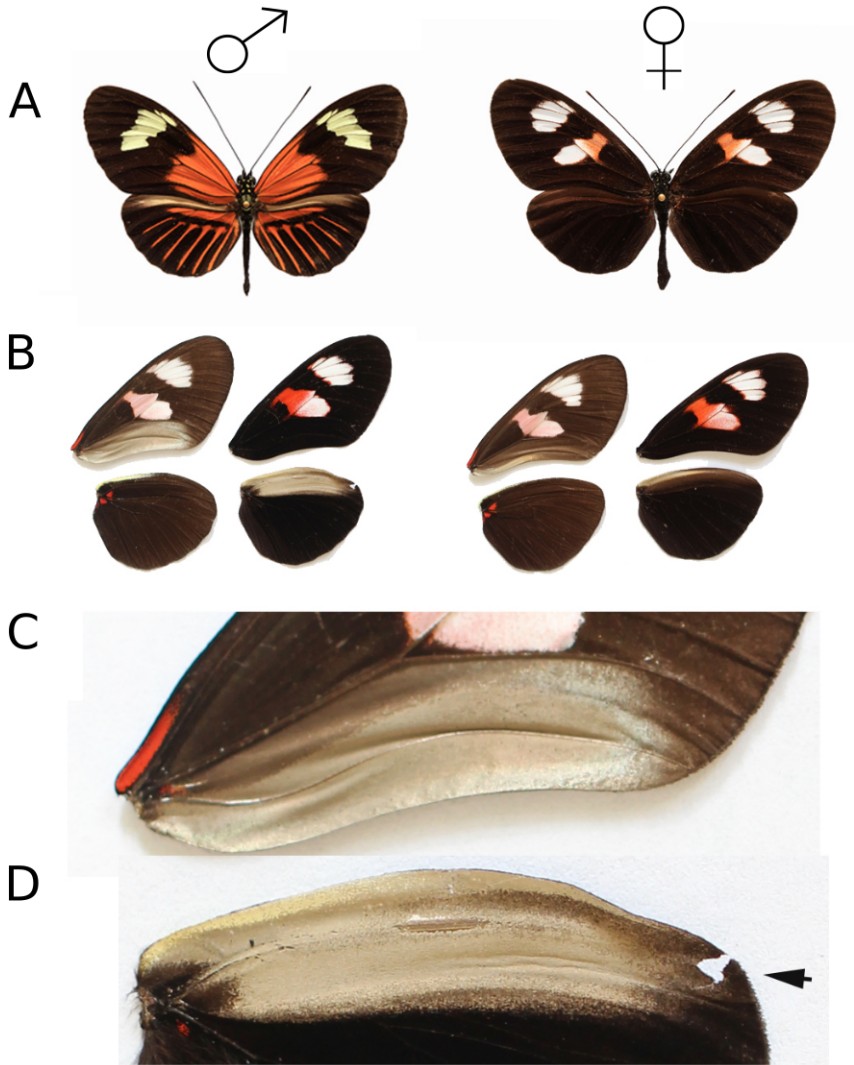

**Figure 1** *Heliconius melpomene* **wings showing androconial dimorphism.** (A) *H. melpomene malleti* (Ecuador sample, left) and *H. melpomene plesseni* (Ecuador sample, right). (B) Dissected wings from specimens of *H. melpomene plesseni* showing sexual dimorphism in the androconial region, with male (left) and female (right). For each sex, the left set of wings shows the ventral surface and the right set the dorsal surface. (C) Expanded view of the male forewing overlapping region. The pale grey-brown region was dissected for chemical analysis. (D) Expanded view of the male hindwing androconial region, with arrow highlight the vein $Sc + R_1$. The pale grey-brown region was dissected for chemical analysis. The ventral side of the forewing is on the top and the dorsal side of the hindwing is on the bottom. The pale grey-brown region in the male wing was dissected for chemical analysis.

Soberania National Park, and San Lorenzo National Park (9°17′N, 79°58′W; elevation 130 m). Larvae were reared on *Passiflora williamsi* and *P. biflora*. Adult butterflies were provided with ∼20% sugar solution with *Psychotria sp., Gurania* sp., and *Psiguiria sp.* as pollen sources. Finally, behavioural experiments were carried out in the mimetic and closely related species *Heliconius melpomene malleti* and *H. timareta florencia* reared at the insectaries of Universidad del Rosario (UR) in La Vega, Colombia. These stocks derived

from wild caught individuals from Sucre, Caqueta (01°48′12″N, 75°39′19″W, elevation 1,200 m). Larvae were reared on *Passiflora oerstedii* and adults were provided with *Psiguiria sp.* as pollen source and ∼20% sugar solution.

## Morphological analysis

The detailed morphology of androconial scales was determined using a Field Emission Scanning Electron Microscope. Three males and two females of *H. melpomene* from Ecuador were used for this analysis. The overlap grey scale region was dissected out from both hind and forewings and attached to aluminium stubs with carbon tabs and subsequently coated with 20 nm of gold using a Quorum/Emitech sputter coater. The gold-coated regions were then viewed in an FEI XL30 FEGSEM operated at 5 kV. Images were recorded digitally using XL30 software at 500× magnification.

## Characterization of potential male sex pheromone

Wing tissue from ten males (five newly emerged and five 10-day old) and five females (10-day old) from the Ecuador stock was collected between November 2011 and March 2012 for chemical analysis. Wings were dissected into four parts: forewing overlap, hindwing androconia, forewing rest and hindwing rest. The 'androconia' and 'overlap' regions corresponded to the grey-brown region (Figs. 1B, 1C and 1D), with rest corresponding to the remaining portion of the wing which is not overlapping. In females, a region corresponding in size and extent to the grey-brown region seen in males was dissected. The dissected sections were then immediately placed in 200 µl hexane or dichloromethane in 2 mL glass vials and allowed to soak for three hours. Initial analysis showed no major differences in extracted chemicals between hexane and dichloromethane extracts (data not shown). Therefore, the more polar dichloromethane was used in later analyses. Due to a larger available stock of *H. melpomene rosina* for behavioural experiments, hindwing androconial tissue was also then collected from 20 males and 11 females (both 10–12 days old) in Panama between February and July 2016. The tissue was soaked in 200 µl dichloromethane in 2 ml glass vials, with PTFE-coated caps, for one hour. The extraction time was shortened as this had no influence on the results (Fig. S1). The solvent was then transferred to new vials and stored at −20 °C. Samples were evaporated under ambient conditions at room temperature prior to analysis.

Mature male androconial extracts from the Ecuador stock were analysed by gas chromatography/mass spectrometry (GC/MS) using a Hewlett-Packard model 5975 mass-selective detector connected to a Hewlett-Packard GC model 7890A, and equipped with a Hewlett-Packard ALS 7683B autosampler. All other Ecuador extracts were analysed by comparison to the male androconial results. Extracts from the Panama stock were analysed by GC/MS using a Hewlett-Packard model 5977 mass-selective detector connected to a Hewlett-Packard GC model 7890B, and equipped with a Hewlett-Packard ALS 7693 autosampler. HP-5MS fused silica capillary columns (30 m × 0.25 mm, 0.25 µm; Agilent, Santa Clara, CA, USA) were used in both GCs. In both cases, injection was performed in splitless mode (250 °C injector temperature) with helium as the carrier gas (constant flow of 1.2 ml/min). The temperature programme started at 50 , was held for 5 min, and then

rose at a rate of 5 °C/min–320 °C, before being held at 320 °C for 5 min. Components were identified by comparison of mass spectra and gas chromatographic Kovats retention index with those of authentic reference samples and also by analysis of mass spectra. The double bond positions of unsaturated compounds were determined by derivatisation with dimethyl disulfide (*Buser et al., 1983*). To confirm chemical structures, alcohols were synthesised from the corresponding methyl esters by reduction according to established procedures (*Becker & Beckert, 1993*, p. 570). The aldehydes were synthesised by oxidation of the respective alcohols (*More & Finney, 2002*).

Compounds found in the extracts were quantified using gas chromatography with flame ionisation detection with a Hewlett-Packard GC model 7890A or 7890B equipped with a Hewlett-Packard ALS 7683B (Ecuador) or 7693 (Panama) autosampler. A BPX-5 fused silica capillary column (SGE, 25 m × 0.22 mm, 0.25 μm) was used in both cases. Injection was performed in splitless mode (250 °C injector temperature) with hydrogen as the carrier gas (constant flow of 1.65 ml/min). The temperature programme started at 50 °C, held for 5 min, and then rose to 320 °C with a heating rate of 5 °C/min. Pentadecyl acetate (10.1 ng) or (*Z*)-4-tridecenyl acetate (1 ng) were used as internal standard for Ecuador samples, and 2-tetradecylacetate (200 ng) for Panama samples. Only compounds eluting earlier than hexacosane were considered for analysis. Later compounds were identified as cuticular hydrocarbons, 2,5-dialkyltetrahydrofurans, cholesterol and artefacts (e.g., phthalates or adipates). The variability in the late eluting cuticular hydrocarbons was low and did not show characteristic differences between samples.

For the Ecuador samples, groups were visualised as boxplots, due to the high frequency of absent compounds in the samples. We then used non-parametric Kruskal–Wallis to test for differences between the amounts of compounds present in different wing regions of mature males, and also between age and sex categories. This was followed up by Dunn post-hoc testing (*Dinno, 2017*; *Ogle, 2017*), with Bonferroni correction.

For the Panama samples, due to the higher sample size, and larger number of compounds identified we visualised the males and females as two groups using a non-metric multidimensional scaling (NMDS ordination, based on a Bray-curtis similarity matrix. We used the metaMDS function in the package vegan (*Oksanen et al., 2017*), with visualisation using the package ade4 (*Dray & Dufour, 2007*). This was followed up with ANOSIM to compare differences between groups, and non-parametric Kruskal–Wallis tests to determine which compounds differed between sexes. All statistical analyses were performed with *R* version 3.3.1 (*R Core Team, 2016*).

### Behavioural experiments

To test female acceptance of male pheromones, behavioural experiments were conducted in insectaries at STRI, Gamboa, Panama between February and July 2016, and also in insectaries at UR in La Vega, Colombia between November 2015 and June 2016. One day old virgin females were presented with a control male and a 'pheromone blocked' male, both of which were at least ten days old. Males from Panama were treated with transparent nail varnish (Revlon Liquid Quick Dry containing cyclomethicone, isopropyl alcohol, ethylhexyl palmitate, mineral oil and fragrance) applied to wings, following

*Costanzo & Monteiro (2007)*. Males from Colombia were treated with transparent nail varnish (Vogue Fantastic containing butyl acetate, ethyl acetate , nitrocellulose, adipic acid, neopentyl glycol, trimellitic anhydride copolymer, isopropyl alcohol, acetyl tributyl citrate, stearalkonium bentonite, styrene, acrylates copolymer, silica benzophenone-1, calcium sodium borosilicate, synthetic fluorphlogopite, polyethylene terephthalate and polyurethane-11). Pheromone blocked males had the dorsal side of their hindwing androconia blocked, whilst control males had the same region on the ventral side of the wing blocked.

Males were randomly marked using a black Sharpie marker with an 'x' on either their left or right wing for identification purposes during the experiment. In Panama, experiments began at 8.30 am and males were left in the cage until 3 pm. During mating, *Heliconius* pairs invariably remain connected for at least an hour and so observations were made every hour to check for matings. If no mating occurred on the first day, this was repeated the next day with the same butterflies. Behavioural observations were recorded for 17 trials with *H. erato demophoon* and 31 trials with *H. melpomene rosina* for the first two hours of the experiment on day one. Observations were divided into one minute intervals, during which both female and male behaviours were recorded. In Colombia, experiments were conducted from 7 am to 1 pm, checking every 30 min for matings. As before, if no mating occurred on the first day, the experiment was repeated the next day with the same butterflies. Female behavioural observations were recorded for 17 trials with *H. timareta florencia* and 18 trials with *H. melpomene malleti* for the first two hours of the experiment on day one. Observations were divided into one minute intervals and were recorded only when a male was actively courting the female. Four female behaviours were recorded: 'Flutter' refers to a high frequency flutter of the wings with a raised abdominal position carried out when another butterfly is in close proximity, which has typically been interpreted as a rejection behaviour (*Klein & Araújo, 2010*; *Jiggins, 2017*). 'Wings open' refers to when the female is alighted with wings open and abdomen raised but without wing fluttering; 'Abdomen up' refers to when the female is alighted with wings closed and abdomen concealed within the wings; 'Fly away' refers to when the female flies away from the male. Of note, 'Flutter' behaviour was only observed when a male was actively courting the female. Male courtships, previously defined as hovering directly over the female (*Klein & Araújo, 2010*), were recorded. Mating outcome results were analysed with binomial tests. We used generalized linear mixed models (GLMMs) with a binomial error distribution and logit link function to test whether females respond differently to control and experimental males. The response variable was derived from trial minutes in which males courted where females performed a particular behaviour ('success') or did not ('failure'). Significance was determined with likelihood ratio tests comparing models with and without male type included as an explanatory variable. Individual female was included as a random effect in all models to avoid pseudoreplication. All statistical analyses were performed with *R* version 3.3.1 (*R Core Team, 2016*), along with the packages ggplot2 (*Wickham, 2009*), car (*Fox & Weisberg, 2011*) and binom (*Dorai-Raj, 2014*).

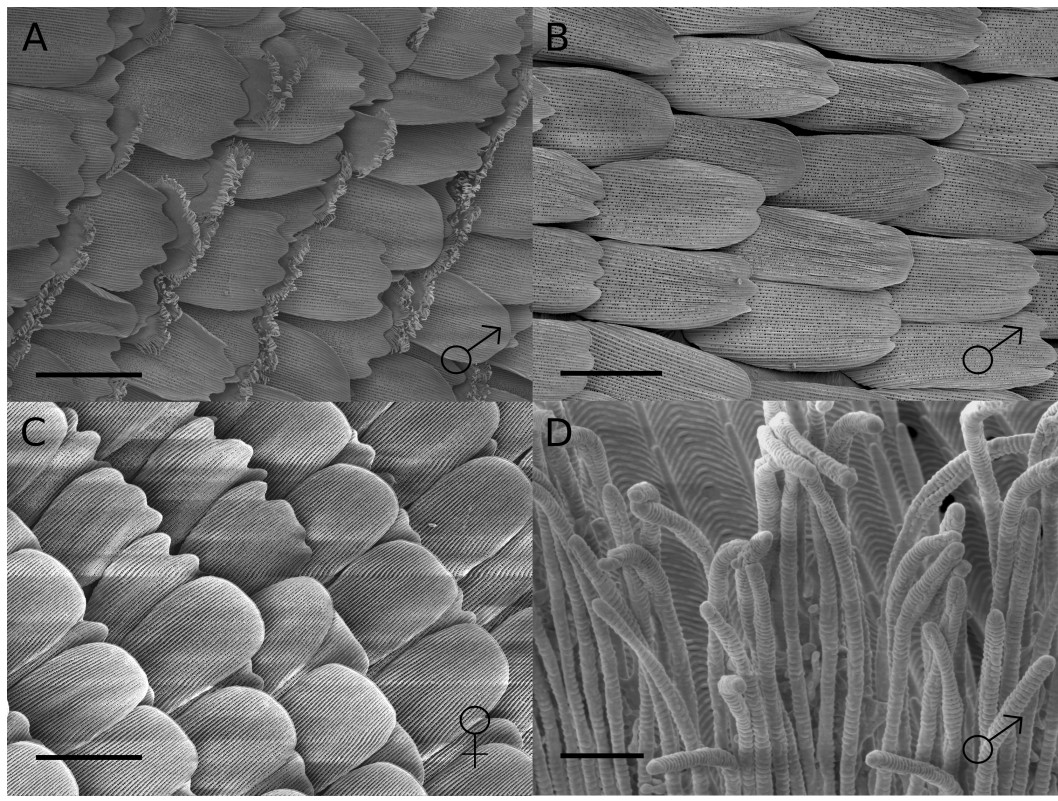

**Figure 2** **SEM images of scales from overlap regions of *H. melpomene* wings.** (A) Male hindwing; (B) male forewing and; (C) female hindwing at 500× magnification. (D) Magnified view of brush-like structures of the special scales in the male hindwing androconial region. Scale bars indicate 50 μm (A–C) and 2 μm (D).

## RESULTS

### Morphological analysis

We identified a marked sexual dimorphism in scale structure (Fig. 2). In the central region of the male hindwing androconia along vein $Sc + R_1$ we identified specialised scales (Fig. 2A), which were absent in females and in the forewing overlap region of males (Figs. 2B and 2C). These scales had brush-like structures at their distal end (Fig. 2D), and were not detected in any other wing region examined. The brush-like scales were found in alternating rows with scales with a normal structure. Moving away from the $Sc + R_1$ wing vein, the density and width of these scales decreased, with isolated brush-like scales found completely surrounded by normal scales. In addition, the base of these brush-like scales was more swollen and glandular as compared to other scales (Fig. 3).

### Characterization of potential male sex pheromone

We initially investigated candidate wing pheromone composition using a stock of butterflies from Ecuador. By use of GC/MS and synthesis, six compounds were consistently found

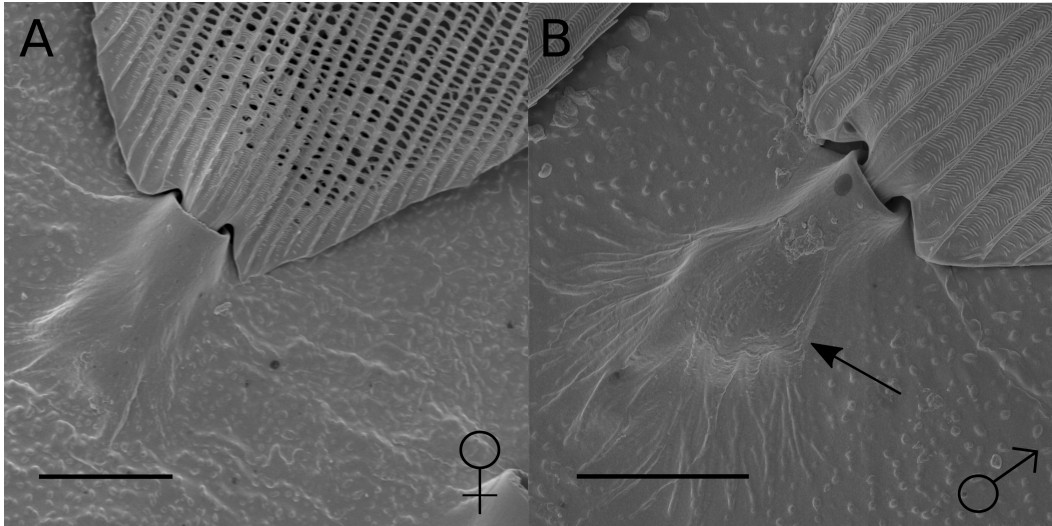

**Figure 3**  **SEM images of scales from hindwing overlap region in female and male *H. melpomene*.** (A) Scale from wing-overlap region of female. (B) Scale from androconial region of male with brush-like structures; the arrow highlights the bulge in the scale base in this region. Scale bars indicate 10 μm.

in the male wing extracts from these samples (Fig. 4) that were identified as the aldehydes $(Z)$-9-octadecenal, octadecanal, $(Z)$-11-icosenal, icosanal, and $(Z)$-13-docosenal and the alkane henicosane ($C_{21}$).

Firstly, a comparison of different wing regions of 10-day old males was carried out (Fig. 5A). Henicosane was found in all regions of the wing and was not considered in further analysis. The amount of $(Z)$-9-octadecenal was not significantly different between area categories. Octadecanal, $(Z)$-11-icosenal, icosanal and $(Z)$-13-docosenal showed significant differences between wing areas. Post-hoc testing found that these four compounds were significantly more abundant in the hindwing androconia than the rest of the forewing and hindwing, but not the forewing overlap region (see Table S1 for statistical details). We suggest that the hindwing overlap region be referred to as the androconial region, based on morphological and chemical analyses. The potential role of the, forewing overlap region as an androconia remains to be demonstrated.

Secondly, the hindwing overlap region of old males, old females, and young males were compared (Fig. 5B). With the exception of henicosane, the other compounds were observed to be age-specific and sex-specific. $(Z)$-9-octadecenal was found more in old males than young males or old females but this was not statistically significant. In contrast, octadecanal, $(Z)$-11-icosenal, icosanal and $(Z)$-13-docosenal showed significant differences between age and sex categories. Post-hoc testing found that these compounds were all present in significantly greater amounts in old males than young males or old females (See Table S2 for statistical details).

As stocks of *H. melpomene rosina* from Panama were available for behavioural assays, we then investigated the chemical composition of this population, using a larger sample size.
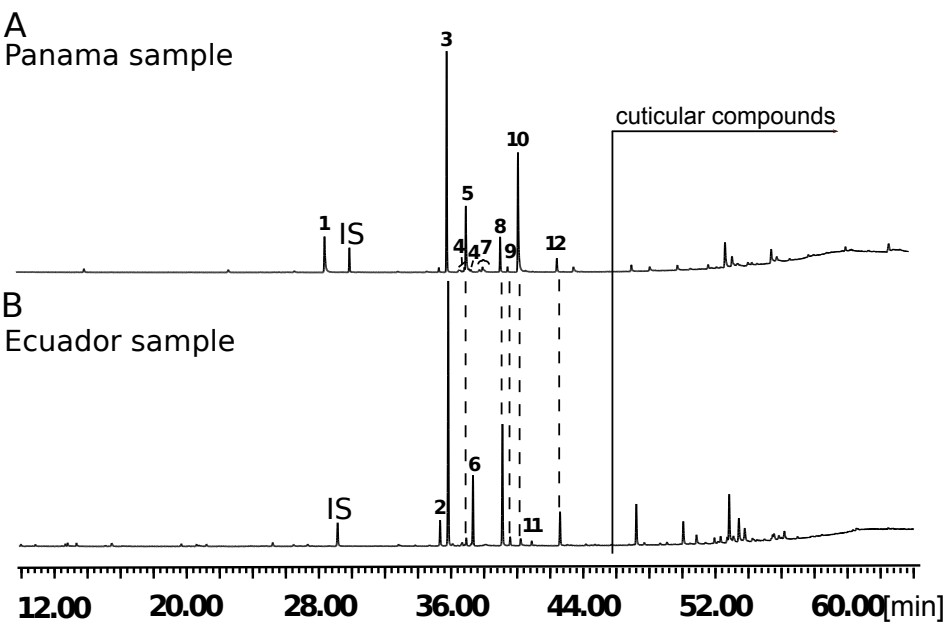

**Figure 4 Regional differences in male androconial extracts.** Total ion chromatogram of extract from the androconial region of (A) a Panamanian *H. melpomene rosina* hindwing and (B) an Ecuadorian *H. melpomene* hindwing. 1, syringaldehyde; 2, (*Z*)-9-octadecenal; 3, octadecanal; 4, methyloctadecanals; 5, 1-octadecanol; 6, henicosane; 7, methyloctadecan-1-ols and nonadecanal; 8, (*Z*)-11-icosenal; 9, icosanal; 10, (*Z*)-11- icosenol; 11, tricosane; 12, (*Z*)-13-docosenal. All peaks eluting later than 44 min are cuticular compounds consisting of larger *n*-alkanes, 2,5-dialkyltetrahydrofurans, cholesterol or are contaminations. IS, internal standard.

These Panama samples showed some similarities to the Ecuadorean samples, although they contained more compounds and in higher amounts (Fig. 4; Fig. S2 and Table S3). Females and males grouped separately with NMDS visualisation, and these groups were significantly different (Fig. S2). In this larger dataset, (*Z*)-9-octadecenal, octadecanal, (*Z*)-11-icosenal, icosanal, (*Z*)-13-docosenal and henicosane were all found in significantly larger amounts in old males than old females, along with many other compounds (Table S3). Small amounts of nonadecanal, methyl-branched octadecanals and their respective alcohols occurred that had not been detected in the Ecuador samples, potentially due to the difference in equipment sensitivity, genuine geographic variation, or the fact that the Ecuadorean butterflies have spent more generations in captivity. Additionally, syringaldehyde was present, which was not detected in the Ecuador samples.

## Behavioural experiments

In our mate choice trials, females of all four species/races discriminated against conspecific males in which pheromone transmission was experimentally blocked (Table 1). Across all four taxa tested, only seven of 71 matings (9.8%) were with the pheromone blocked male, with the remaining 64 matings (90%) being with the control (unblocked) males. This was not due to altered male courtship attempts as control and experimental males courted equally in three out of four species (Fig. S3). In experiments with *H. timareta*

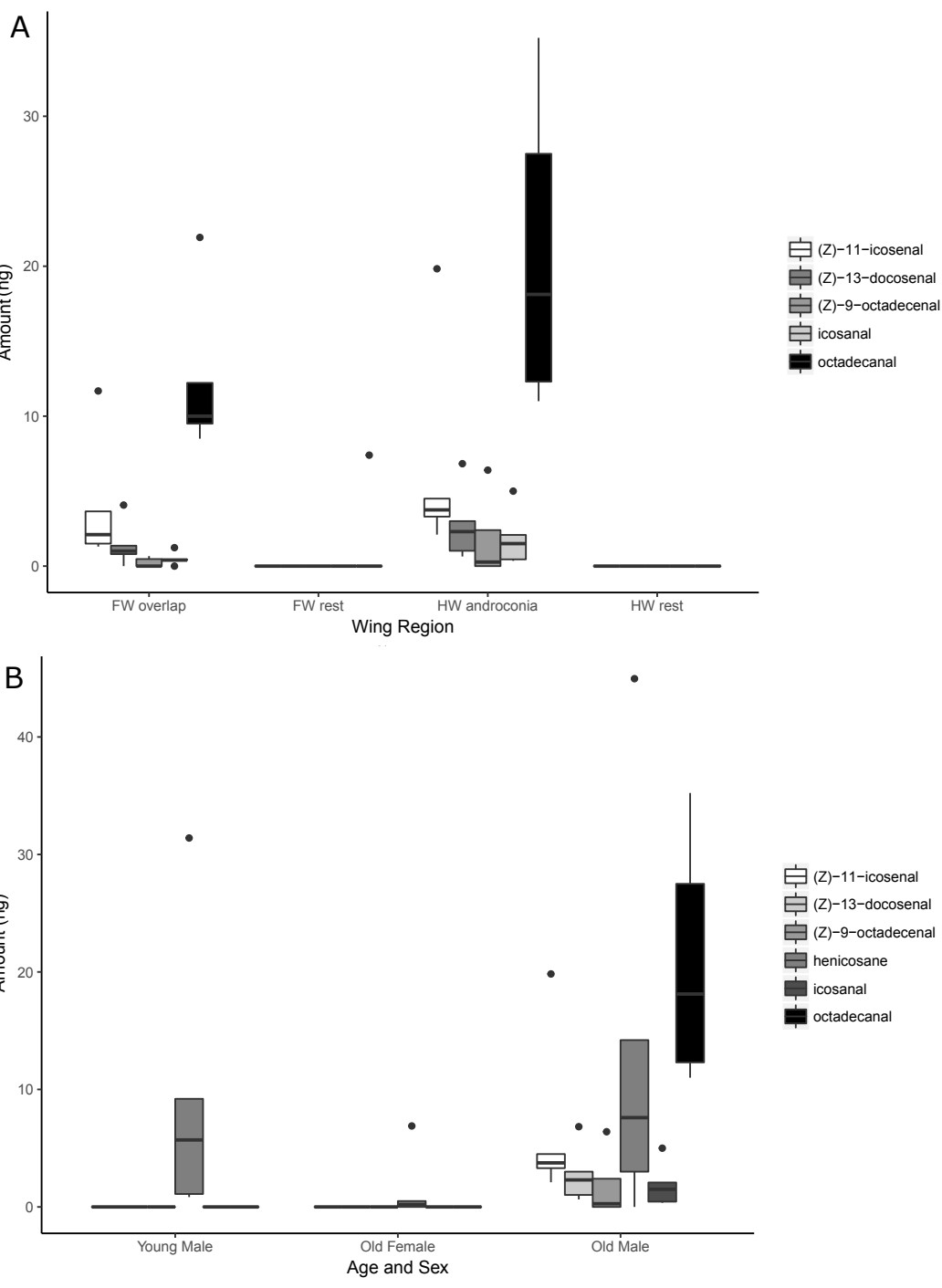

**Figure 5** **Compounds detected by GC/MS of *H. melpomene* (Ecuador samples) wing extracts.** (A) Presence of compounds in different wing regions of five males (10 days post-eclosion). (B) Presence of compounds in five females (10 days post-eclosion), five young males (0 days post-eclosion) and five old males (10 days post-eclosion).

**Table 1  Outcome of mate choice trials across different species/races.** Proportion of successful copulations with the control male was tested using an exact binomial test. Females mated significantly more with the control male than the experimental (pheromone-blocked) male in all four populations. Statistical analysis based on females which mated.

| Species | Mated with control | Mated with experimental | Did not mate | *p*-value (exact binomial test) |
|---|---|---|---|---|
| *H. melpomene rosina* | 15 | 0 | 18 | <0.001 |
| *H. erato demophoon* | 14 | 1 | 31 | <0.001 |
| *H. melpomene malleti* | 19 | 3 | 8 | <0.001 |
| *H. timareta florencia* | 16 | 3 | 5 | <0.01 |

*florencia*, the control males courted more than experimental males (Fig. S3). When data from *H. timareta florencia* are excluded, 48 out of 52 matings (92%) occurred with the control (unblocked) males.

We observed no consistent significant differences in female behavioural responses towards control and experimental males (Fig. 6). Some female behaviours were observed more often towards experimental males in experiments with *H. melpomene malleti* and *H. timareta florencia* (Fig. 6). In particular, *H. melpomene malleti* females were more likely to open their wings towards experimental males ($2\Delta\ln L = 17.093$, d.f. $= 1$, $p < 0.001$), fly away ($2\Delta\ln L = 8.0356$, d.f. $= 1$, $p < 0.01$) and also flutter ($2\Delta\ln L = 15.823$, d.f. $= 1$, $p < 0.001$). Similarly, *H. timareta florencia* females were also more likely to open their wings towards experimental males ($2\Delta\ln L = 22.909$, d.f. $= 1$, $p < 0.001$), fly away ($2\Delta\ln L = 6.1368$, d.f. $= 1$, $p < 0.001$), and flutter ($2\Delta\ln L = 26.037$, d.f. $= 1$, $p < 0.001$). To ensure that behavioural trials without successful mating were not skewing our analysis of female behaviours, we additionally analysed differences between males that mated with the females versus those that did not mate (including those from experiments without matings). Differences between female behaviour towards mated versus unmated males did not differ from differences seen in behaviour towards experimental versus control males (Fig. S4). However, despite these differences, these behavioural responses were not consistently observed across the four species/races tested. Some of these results are driven by just a few individual females, with differences in behaviour no longer significant when they are removed. Furthermore, although some significant female behavioural responses were seen for *H. melpomene malleti* in Colombia, no corresponding difference was found for *H. melpomene rosina* in Panama, where a larger number of courtships were observed, so caution should be taken when interpreting these results.

## DISCUSSION

Visual cues are known to be important for mate finding and courtship behaviours by male *Heliconius* butterflies, with implications for reproductive isolation and speciation (*Merrill et al., 2015*). Here, we have shown that female choice based on chemical signalling is also important for reproduction. We have identified compounds associated with sexually mature male wings and described morphological structures putatively involved in pheromone release. Furthermore, we have shown that chemical signalling is involved in mating in

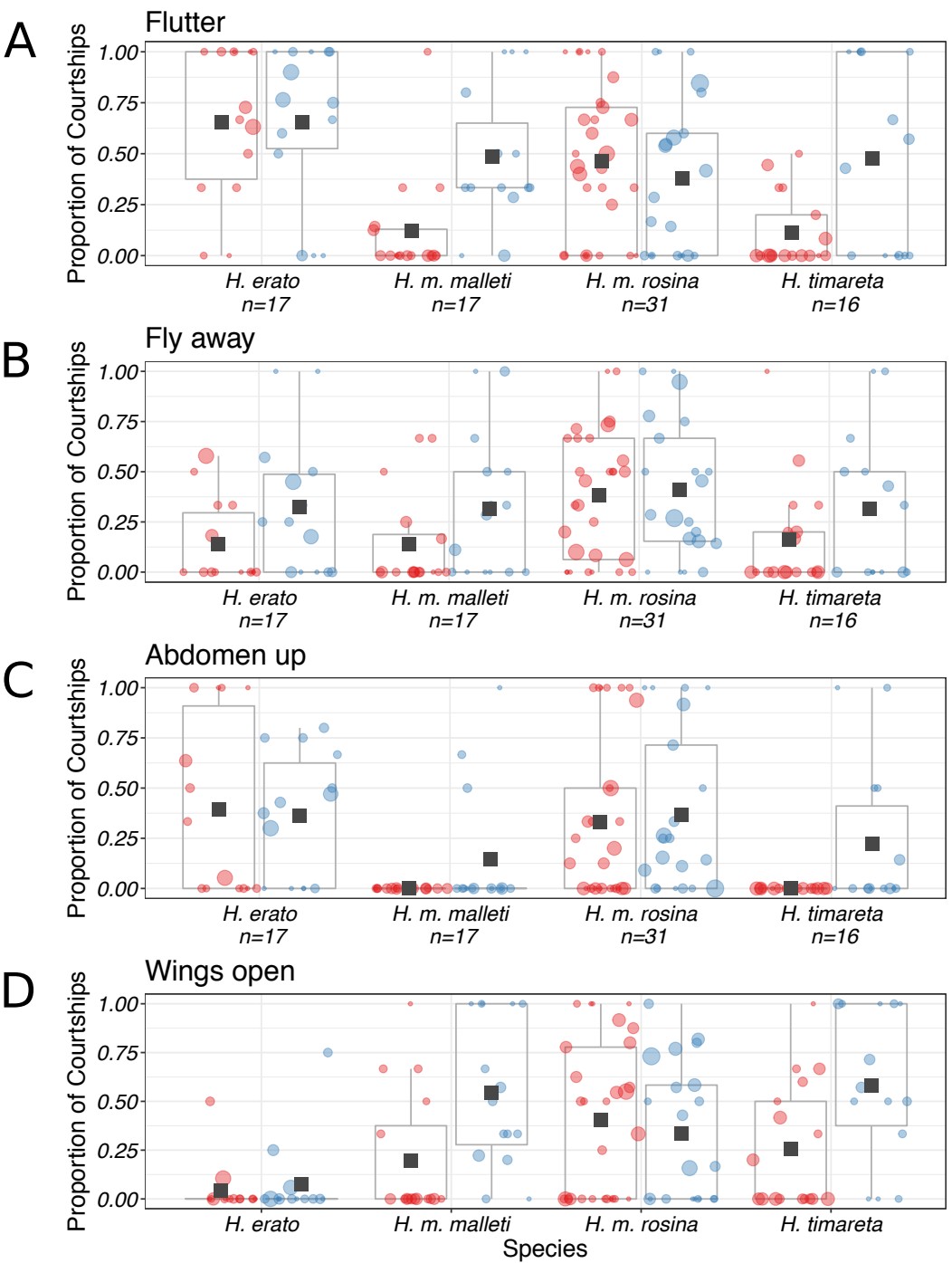

**Figure 6 Proportion of courtships which resulted in different female behavioural responses.** Control males are represented in red (left) and experimental males in blue (right). Means are marked with a black square and boxplots mark the inter-quartile ranges. Size of datapoint is proportional to the number of courtships by that male. Female behavioural responses (A) Flutter; (B) Fly away; (C) Abdomen up; (D) Wings open.

*Heliconius*, with females from three different species across four races showing strong discrimination against males which have had their androconia experimentally blocked.

Our results are broadly comparable with another recent analysis of wing compounds in *Heliconius* (*Mérot et al., 2015*), although the previous study did not compare different wing regions, or males and females of the same age. As the previous study also did not use synthesis to identify compounds, our work is highly complementary and extends their results to confirm region- and age-specific localization of compounds to older male androconia. Male *Heliconius* do not become sexually active until several days after eclosion, so the absence of these compounds from females and younger males is strongly suggestive of a role in mating behaviour. Future experiments will be required to determine if these compounds are sequestered from larval host plants, and if there is genetic control of the production of these compounds, both of which could facilitate a role in reproductive isolation.

The restriction of these five putative male sex pheromones, $(Z)$-9-octadecenal, octadecanal, $(Z)$-11-icosenal, icosanal, and $(Z)$-13-docosenal, to the hindwing androconia of mature males (Fig. 5A) suggests that pheromone storage or production is restricted to the hindwing. This is supported by the scanning electron microscope images which show special brush-like scales in the androconial region (Fig. 2A), located primarily around and along the hindwing vein $Sc + R_1$, similar to the depiction in Fig. 73 of Emsley's previous morphological analysis (*Emsley, 1963*). Similar scales have been described from light microscopy in other *Heliconius* species, but not previously in *H. melpomene* (*Müller, 1912*; *Barth, 1952*). The base of these special brush-like scales was more swollen and glandular as compared to other scales (Fig. 3), perhaps indicating a role in storage or production of pheromones by these scales. Trace amounts of chemicals on the forewing overlap region may be due to contact in the overlapping portion of the fore- and hindwings, and both wings may play a role in dispersal of the compounds during courtship.

Samples from Panama showed both a greater diversity and amounts of compounds (Fig. S2 and Table S3). This might reflect an issue with inbreeding in the Ecuador population because they were obtained from a commercial breeder, or technical differences between the two locations where the analysis was performed. However, it could also reflect differences in rearing conditions or genuine variation between geographic populations of *H. melpomene*. Further work will be needed to confirm the nature and extent of geographic and individual variation in pheromone composition.

Females exhibited a strong preference for males which did not have their androconia blocked. This suggests that, as in other butterfly systems (*Costanzo & Monteiro, 2007*), female *Heliconius* are actively involved in mating decisions. Nonetheless, there were no consistent differences in the female behaviours we recorded in our experiments. It is possible that the important female preference behaviours are subtle and were missed in our study, or that our sample size may be too small to detect behavioural differences due to individual variation. Alternatively, female acceptance of a male may instead simply represent a decision to stop rejection behaviours, and therefore not be associated with any particular characteristic behavioural response.

It remains unclear which compounds are biologically active and exactly what information is being conveyed. The signal clearly influences female mating decisions, and these compounds may convey complex information about male species identity, quality, age etc. that are interpreted by females. It is also unknown whether females, like males, use visual cues in courtship. The use of multiple signals is common in animal communication (*Candolin, 2003*). Whilst butterflies primarily use visual cues to locate mates (*Kemp & Rutowski, 2011*), it has been shown in *B. anynana* that in addition to visual cues, chemical cues also play a role and are equally important in sexual selection by female choice (*Costanzo & Monteiro, 2007*). Our work establishes the potential for similar multimodal signalling in *Heliconius* butterflies.

This study provides evidence for the importance of pheromones in intraspecific mate choice in *Heliconius* butterflies. Evolution of cues within populations could lead to reproductive isolation between populations if both cues, and their corresponding preferences, diverge (*Ptacek, 2000*). In other butterflies, male wing compounds contribute to reproductive isolation between closely related species (*Grula, McChesney & Taylor, 1980*; *Phelan & Baker, 1987*; *Bacquet et al., 2015*). Evidence suggests that strong pre-mating barriers in addition to mate preference based on colour wing pattern exist between *Heliconius cydno* and *H. melpomene* (which differ in colour pattern) and between the latter and *H. timareta* (which are mimetic) (*Mérot et al., 2017*; *Giraldo et al., 2008*). For example, *H. cydno* males show a preference for their own pattern over that of the closely related *H. melpomene*, but will court wing pattern models of *H. melpomene*. *Heliconius cydno* males, however, have virtually never been observed mating with *H. melpomene* females (*Naisbit, Jiggins & Mallet, 2001*). On the other hand, although males of the mimetic *H. timareta florencia* and *H. melpomene malleti* equally court female wing models of both species, interspecific matings occur in very low frequency (*Sánchez et al., 2015*; *Mérot et al., 2017*). Furthermore, male androconial compounds differ between species (*Mérot et al., 2015*; *Mann et al., 2017*), suggesting that they could play a role in reproductive isolation. Future work will allow us to explore the multidimensional aspect of speciation by understanding both male and female choice, and the role that multiple modes of signalling could play in reproductive isolation.

## ACKNOWLEDGEMENTS

We acknowledge the support of Caroline Nieberding and Christer Löfstedt who provided early encouragement to pursue this project, and Kelsey Byers who gave useful advice on data analysis and manuscript preparation. We also thank Andrew Mongue and an anonymous reviewer who provided helpful comments on the manuscript. We thank our team at the insectaries in Panama, including Oscar Paneso, Sylvia Fernanda Garza Reyes and Rachel Crisp. We also thank Oscar Penagos for his technical support breeding butterflies at insectaries in La Vega, Colombia.

### Funding

KD is funded by a Natural Environment Research Council Doctoral Training Partnership. SJ was funded by a Manmohan Singh studentship from St John's College. RMM was funded by a Junior Research Fellowship at King's College, Cambridge. CDJ and RMM are supported by a European Research Council grant number 339873 Speciation Genetics. MFG, CS and CPD were funded by the Universidad del Rosario FIUR grant QDN-DG001 and COLCIENCIAS (Grant FP44842-5-2017). WOM was supported by the Smithsonian Tropical Research Institute and NSF grant DEB 1257689. CRM was supported by a STRI Predoctoral Fellowship. The funders had no role in study design, data collection and analysis, decision to publish, or preparation of the manuscript.

### Grant Disclosures

The following grant information was disclosed by the authors:
Natural Environment Research Council.
Doctoral Training Partnership.
Manmohan Singh studentship.
Junior Research Fellowship.
European Research Council: 339873.
Universidad del Rosario FIUR: QDN-DG001.
COLCIENCIAS: FP44842-5-2017.
Smithsonian Tropical Research Institute and NSF: DEB 1257689.

### Competing Interests

The authors declare there are no competing interests.

### Author Contributions

- Kathy Darragh and Sohini Vanjari conceived and designed the experiments, performed the experiments, analyzed the data, contributed reagents/materials/analysis tools, wrote the paper, prepared figures and/or tables, reviewed drafts of the paper.
- Florian Mann conceived and designed the experiments, performed the experiments, analyzed the data, contributed reagents/materials/analysis tools, prepared figures and/or tables, reviewed drafts of the paper.
- Maria F. Gonzalez-Rojas conceived and designed the experiments, performed the experiments, contributed reagents/materials/analysis tools, reviewed drafts of the paper.
- Colin R. Morrison performed the experiments, reviewed drafts of the paper.
- Camilo Salazar, Carolina Pardo-Diaz and W. Owen McMillan conceived and designed the experiments, contributed reagents/materials/analysis tools, reviewed drafts of the paper.
- Richard M. Merrill conceived and designed the experiments, analyzed the data, contributed reagents/materials/analysis tools, wrote the paper, reviewed drafts of the paper.

- Stefan Schulz conceived and designed the experiments, analyzed the data, contributed reagents/materials/analysis tools, reviewed drafts of the paper.
- Chris D. Jiggins conceived and designed the experiments, analyzed the data, contributed reagents/materials/analysis tools, wrote the paper, prepared figures and/or tables, reviewed drafts of the paper.

## Data Availability

Raw data can be found in the Supplemental Information.

## Supplemental Information

Supplemental information for this article can be found online at http://dx.doi.org/10.7717/peerj.3953#supplemental-information.

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
