# Peer review of "Male sex pheromone components in Heliconius butterflies released by the androconia affect female choice"

_PeerJ, doi:10.7717/peerj.3953_

## Round 0.1 · original submission · Major Revisions

· Academic Editor

Major Revisions

My name is Christopher Cutler and I have been contacted by PeerJ staff to handle the submission as you are appealing the Reject editorial decision made by the prior Academic Editor. 
I am issuing this “revised” decision so you can upload the most recent version of your manuscript and a point by point response to the reviewers.

· Appeal

Appeal

Dear Sophie

I am writing on behalf of my PhD student Kathy in response to your email - see below. We are quite puzzled as we had independently approached the Academic Editor, Ann Hedrick, and she had encouraged us to resubmit a revised version via the editorial office. The version submitted by Kathy included a detailed response to referees and rebuttal. This has involved a lot of work by Kathy who is currently away on fieldwork in Panama, so your email was something of a disheartening experience.

Anyway, we would like to proceed with an appeal if thats possible. The submission already includes a rebuttal and tracked changes - please could you let us know if we need to do anything else

Best

Chris


· · Academic Editor

Reject

Unfortunately, the reviewers raised some very serious concerns, summarized here:

1) The large number of failed choice assays in two of the populations (H. melpomene rosina and H. erato demophoon). The authors appear to have used observational data for a portion of the females who did not mate with either the control or the experimental male (and thus have not behaviorally exhibited a clear preference) to assess whether females behave differently around preferred or non-preferred males. However, as these females did not mate with either male, it is unknown which male they prefer, therefore their data may confound the results.

2) The small sample sizes used for the behavioral assays, given that behavioral data tends to include a large amount of individual variance. The sample sizes are fine for the binary mating outcome analyses, but may be too small to detect biologically relevant female behaviors when searching for female behavior specifically focused towards attractive or less attractive males. The authors over interpret these findings in the discussion.

3) 3 different subspecies of H. melpomene from different geographic locations and environmental rearing procedures were used more or less interchangeably for morphological, chemical and behavioural experiments. Given that the morphological chemical and behavioural experiments were each done with a different (mix?) of subspecies, and that subspecies in Lepidoptera are well known to differ adaptively in sex pheromone communication, it is very hard to make conclusions from one experiment in relation with another, e.g. for example whether the chemical distinctiveness of mature male androconial areas of one subspecies may explain the female differential behavior in presence of male wing extracts (of another subspecies).

Thus most of the conclusions of this work hence remain subject to further experimental confirmation.

For these reasons, I must reject your manuscript for publication. I wish you luck in finding another venue.

·

Basic reporting

This manuscript is well written with a solid foundation of literature to motivate the research conducted within; figures are legible and appropriately explained in captions. I have no major concerns with the analyses or writing, but please go through the references for consistency in formatting, especially italicizing genus and species names.

Experimental design

The authors do a thorough job of investigating male pheromone production in three distinct ways, strengthening the conclusions to be drawn from their findings. Reporting of protocols is explicit and clear from data collection to software used in statistical analyses.

Validity of the findings

The authors present strong evidence for the existence of pheromones in mature males that enhance mating success. Results from the structural, chemical, and behavioral experiments work together to show that chemicals produced (or at least stored) in the hindwings of mature males contribute to reproductive success. I have only a few comments about the reporting of these results and their implications:
First, in L 333-337, the authors note that courting rates differed between control and treated males in one of the four species, yet they report percentages for successful matings pooling all tested species. What do the percentages of mating success look like when excluding the species in which control males courted more frequently than experimentally treated males?
Next, experimental addition of pheromones is the other side of the coin to experimental removal/blocking of such compounds. In L 422-424, the authors give the example of H. cydno males unsuccessfully courting heterospecific females. Assuming that the pheromone composition of H. cydno is made up of a handful of similar compounds to those studied here, how difficult would it be to synthesize and experimentally apply such compounds to these males to do a complimentary follow up experiment?
Finally, throughout this manuscript, the male butterflies are said to have both forewing and hindwing androconia. This is a reasonable assessment based on morphology alone, as Figure 1 shows sexual dimorphism in both wing pairs. However, the chemical analyses of different parts of the wings suggest that it is largely if not entirely the hindwing region that produces/stores pheromones. Furthermore, the behavioral studies only blocked regions of the hindwing manipulated males to demonstrate differential mating success. It seems that the dorsal hindwing is the key area for pheromone signals in these species. In light of these results, is it less appropriate to continue calling the forewing regions androconia? Perhaps a sentence or two could be added about if/how results presented here affect the classification of the regions on the forewing?

Additional comments

This manuscript and the research within are both of high quality and definitely suitable for publication. As you can see, my corrections and suggestions are all minor. At most, a couple of points could be clarified with another line or two of explanation in the text, but no additional analyses will be required. I have mentioned some suggestions above with specific line references, but please see the attached file for the full list of line-by-line comments.

Reviewer 2 ·

Basic reporting

In general, the authors use clear, unambiguous, and professional language throughout. Their introduction and background are sufficient, the literature is well referenced and relevant, and the article structure conforms to discipline norm.

The figures are relevant and of high quality, though S3 is highly relevant to the main text of the article, and should be included in the main text.

The raw data was also supplied.

Experimental design

The authors posed a study searching for the presence of androchonia in four populations of Heliconius butterflies, and tested whether the presence of the chemicals produced by these androchonia influence male mating success. This is a highly relevant and timely research question for speciation and mate preference work, as Heliconius butterflies are a model system for speciation research. Male preference for female wing color is heavily studied in this system, but, to date, very little is known about either female preference or the use of chemical signaling in Heliconius mating decisions. Therefore, this study has the potential to substantially broaden the scope of speciation research in one of the most well used study systems for reproductive isolation and population genetics research.

The characterization of the chemical compounds associated with the proposed androchonia is outside of my technical area of expertise, therefore I suggest someone else review those methods. However, they appeared clear to me, and detailed enough that I believe they could be easily replicated.

I do, however, have a few comments and concerns about the experimental design of the behavioral assays.

1) Given that the authors only recorded female behavior when males were courting, the authors should describe (or cite a source describing) male courtship in H. melpomene, H. erato, and H. timareta. Male courtship is a very important part of this study, and it is currently undefined. Please include.

2) I am concerned by the large number of failed choice assays in two of the populations (H. melpomene rosina and H. erato demophoon). The authors appear to have used observational data for a portion of the females who did not mate with either the control or the experimental male (and thus have not behaviorally exhibited a clear preference) to assess whether females behave differently around preferred or non-preferred males. However, as these females did not mate with either male, it is unknown which male they prefer, therefore their data may confound the results.

3) I am also slightly concerned about the small sample sizes used for the behavioral assays, given that behavioral data tends to include a large amount of individual variance. The sample sizes are fine for the binary mating outcome analyses, but may be too small to detect biologically relevant female behaviors when searching for female behavior specifically focused towards attractive or less attractive males. The authors do mention these low sample sizes in the results section, but they then over interpret these findings in the discussion.

Validity of the findings

The authors did a good job presenting their work in the context of published research on the role of chemical signaling in butterfly mate choice. Their histology and chemical characterization results are clear, and their mate choice study, though with small sample size, has clear results.

I am, however, concerned by the selection of which butterflies were used for further analysis of female behavior (which I describe in more detail in the general comments below), and the authors classification of these behaviors as rejection behaviors.

The authors describe all female behavior as rejection behavior, however they do not have supporting evidence that these behaviors are, in fact, rejection behaviors in Heliconius melpomene, Heliconius erato, or Heliconius timareta. The authors should take an unbiased approach when describing the female behaviors, and simply state that they observed and recorded four different female behaviors: flutter, wings open, abdomen lifted, and flying away. Unless there is another study in H. melpomene, H. erato, and H. timareta that I am not aware of which clearly shows that fluttering, wing opening, abdomen lifting, and flight happen at a higher frequency when males are courting females than when females are not being courted by males? Fluttering and wing opening are fairly common butterfly behaviors, thus the authors should justify their description of these behaviors as rejection behaviors, or use more neutral language.

Additional comments

The authors integrated an interesting set of chemical ecology and animal behavior studies to assess whether chemical signals on the wings of male Heliconius butterflies influence male mating success. In general, this is a well written paper describing a timely set of experiments. However, I have a couple of major concerns regarding the behavioral experiment, as well as concerns about data interpretation, which I detail below.


1) For both H. melpomene rosina and H. erato demophoon, most of the females did not mate during the two-day choice assay (as illustrated in Table 1). This suggests to me that there may have been something wrong with the set-up. Are the authors sure that the females were old enough to mate? Are the authors sure that the males were old enough to mate? While it is clear that the females that did mate preferred control to androchonia blocked males, I am a bit concerned by the shear number of failed mate choice trials. Particularly given the authors interpretation of female behavior during the observed portion of the choice trials.

Given that most H.m. rosina females did not mate, I find it unsurprising that the authors did not observe a difference in H. m. rosina female behavior around control or androchonia blocked males (as illustrated in Figure S3). In fact, the authors’ behavioral results, coupled with their finding that over half of the H. m. rosina females failed to mate with either the control or the androchonia blocked males suggests that either these females were not reproductive, or that the females did not find either male attractive. Either way, one would expect these females (who did not mate with either male) to treat both male forms equally. I suggest that the authors remove the behavioral data for the females that did not mate, and only compare the behavior of the females with known preferences (ie mating outcomes). Otherwise the authors are using their assumptions about female preference to define behaviors as rejection behaviors, instead of using actual preference data to assess whether females treat males they prefer differently from males they do not prefer.

2) The authors state that the absence of these chemical compounds in young males and older females suggests that the compounds are unlikely to have been sequestered from plants, but that logic seems a bit flawed to me (Lines 370-373). The compounds could be sequestered from plants in males but not females, and only be released in older males. In order to state that the compounds are unlikely to be obtained from direct sequestration of compounds from larval host plants, the authors need to show that males reared on different host plants produce the same chemical compounds.

This does not mean that I do not agree with the authors’ argument that these chemical compounds are important for mate acquisition. I think the authors have adequately shown that these chemical signals increase the likelihood of female acceptance. However, I do not think the authors have sufficient support for their statement in line 372 that there could be genetic control of the production of these compounds. Nor does there have to be genetic control of the production of these compounds for them to play a role in reproductive isolation- if the chemicals were sequestered from host plants, host plant differentiation could be associated with male chemical differentiation and consequent reproductive isolation.

Lines 394-401: I am confused as to why the authors expect females from different species and populations to exhibit the same behaviors when interacting with males. The authors found that females did exhibit population specific behavioral differences in response to preferred versus non-preferred males- why not discuss this further? This is a particularly interesting finding, given the small sample size (which the authors do discuss in their results). As I mentioned earlier, since the authors have not yet shown that these behaviors are, in fact, rejection behaviors (and their results here suggest otherwise), the authors should use a more neutral term to describe the documented female behaviors.

Minor Comments:

Line 193- the authors state that shortening extraction time did not influence results, but do not show the data or cite a paper to support this statement. This should be a relatively straightforward figure or citation. Please include.


Line 425-426: The authors state that male androconial compounds differ between H. timareta and H. melpomene, but this data appears to be unpublished. Is it currently in review? Will it be published soon? Can the authors support this statement with published data?

Reviewer 3 ·

Basic reporting

XX

Experimental design

XX

Validity of the findings

XX

Additional comments

XX

---

## Round 0.2 · accepted · Accept

· Academic Editor

Accept

The referees and I are fully satisfied with the changes you have made to the paper, and your clear responses to previous reviewer concerns. I am pleased to make the final decision on your Appeal and to accept your paper for publication in PeerJ.

Reviewer 2 ·

Basic reporting

The authors have addressed my concerns in their revision.

Experimental design

no comment.

Validity of the findings

The authors have satisfactorily addressed my previous concerns.

Additional comments

I like this study. It is integrative and timely.

I appreciate the revisions the authors made to the text. Particularly their inclusion of separate analyses of the female behavior in successful and unsuccessful mating trials, the detailed descriptions of the behaviors of the butterflies, and their moving the figure of female behavior from supplemental to the main text.

The authors have satisfactorily addressed my previous concerns.

·

Basic reporting

I have only seen the revised version with responses to the previous reviewers. In my eyes this paper complies to all criteria, it is well written, well illustrated and both introduction and discussion are sound.

Experimental design

The authors have convincingly responded to the concerns raised by the previous reviewers and I have no additional comments concerning the experimental design.

Validity of the findings

The authors have perfectly responded to the concerns raised by the previous reviewers, and reformulated parts where a slight over-interpretation might have occurred. No additional comments from my side.

Additional comments

This paper accumulates an important amount of work on male sex pheromones and their role in mating in different Heliconius species/races. It is evident that certain behavioural data are difficult to obtain in rare species/races. But this does not at all reduce the value of the presented data.